# Effect of a Decellularized Omentum Scaffold with Combination of Mesenchymal Stem Cells and Platelet-Rich Plasma on Healing of Critical-Sized Bone Defect: A Rat Model

**Abdulsamet Emet** [1,*] **, Erdi Ozdemir** [1] **, Duygu Uckan Cetinkaya** [2] **, Emine Kilic** [2] **, Ramin Hashemihesar** [3] **, Ali Celalettin Sinan Yuruker** [4] **and Egemen Turhan** [1]

1    Department of Orthopedics and Traumatology, Faculty of Medicine, Hacettepe University, Ankara 06230, Turkey; erdiozdemir@gmail.com (E.O.); egemen.turhan@gmail.com (E.T.)
2    Department of Stem Cell Research and Development, Faculty of Medicine, Hacettepe University, Ankara 06230, Turkey; duyguc2008@gmail.com (D.U.C.); emine.ekilic@gmail.com (E.K.)
3    Department of Histology and Embryology, Istanbul Health and Technology University, Istanbul 34010, Turkey; ramin.hashemihesar@gmail.com
4    Department of Histology and Embryology, Faculty of Medicine, Usak University, Uşak 64000, Turkey; sinany@gmail.com
*    Correspondence: drsametemet@gmail.com; Tel.: +90-537-452-6116

**Abstract:** The high costs and extensive time needed for the treatment of critical-sized bone defects are still major clinical concerns in orthopedic surgery; therefore, researchers continue to look for more cost and time-effective methods. This study aims to investigate the effects of a decellularized omentum scaffold with a combination of platelet-rich plasma (PRP) and mesenchymal stem cells on the healing of critical-sized bone defects. Wistar albino rats ($n$ = 30) were investigated in five groups. Critical-sized bone defects were formed on bilateral radius shafts. No scaffold, decellularized omentum, omentum with PRP and omentum + mesenchymal stem cells was used in group 1 (control group), 2, 3 and 4, respectively. In addition, omentum with a combination of mesenchymal stem cells +PRP was used in group 5. After 6 weeks, both radiological and histological healing were evaluated comparatively among the groups. After the use of a decellularized omentum scaffold, vitality of new cells was maintained, and new bone formation occurred. When compared to the control group, radiological healing was significantly better ($p$ = 0.047) in the omentum and omentum + PRP-treated groups. Furthermore, histological healing was better in the omentum and omentum + PRP-treated groups than the control group ($p$ = 0.001). The use of a decellularized omentum scaffold is suitable in the healing of critical bone defects.

**Keywords:** segmental bone defect; PRP; omentum; scaffold; mesenchymal stem cell

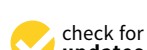



## 1. Introduction

The frequency of bone defect problems encountered by orthopedic surgeons is increasing. High energy trauma, open fractures, bone tumors, infections, debridement of bone for any reason, metabolic diseases and massive osteolysis are the most common causes of bone defects [1]. The main problems in dealing with bone defects are the high costs and extensive time needed for healing. Critical bone defects are difficult situations, and spontaneous healing is not expected despite adequate surgical stabilization [2]. In addition to appropriate surgical stabilization, critical bone defect treatments must include proper and functional segmental replacement regardless of the cause. In the literature, autologous bone grafts (ABG) are described as the gold standard among current treatment modalities [3]. However, due to some disadvantages, such as limited availability and donor site morbidity, the efforts to find more appropriate and optimal scaffold options continue.

The bone scaffold is a 3D matrix that allows osteogenic cell adhesion and proliferation, as well as stimulates cells [4]. Natural and synthetic polymers, ceramic-based scaffolds,

metallic, and composite scaffolds have been tried so far, but none of the used scaffolds contain all the properties of an optimal scaffold [5,6]. Moreover, synthetic scaffolds are associated with rejection, allergic reactions and failure to integrate with host tissues. As a result, tissue engineering in current technological upgrades is seminal in bone defect treatment. Presently, natural extracellular matrix composite scaffolds are of interest because these are accepted as most similar to the original tissue, and they allow cellular adhesion and proliferation. In addition, the main advantage of this scaffold type is being non-antigenic to the native tissue [7].

For these reasons, tissue-engineered adipose substitutes have been developed that promote regeneration rather than repair [8]. The omentum has been used for defects of other organs because of its rich extracellular matrix, ability to adhere to local tissues and growth factors containment, but it is generally used in reconstructive and cardiovascular surgery as a decellularized omentum scaffold. However, in orthopedic surgery, decellularized omentum scaffolds have not been used to date [8].

Successful bone repair is based on a combination of a suitable medium, stem cells and growth factors. Platelet-rich plasma (PRP), which has osteoinductive and osteoconductive properties, is a good candidate to be added to a scaffold [9]. It is proven to have a number of growth and differentiation factors that play important roles in the intracellular signaling pathway for cellular proliferation, osteoid and extracellular matrix formation, and also callus formation in fracture healing [10]. In addition, mesenchymal stem cells (MSCs) were shown to be involved in callus formation [11]. Both direct involvements in callus formation and inflammatory cytokines, such as transforming growth factor (TGF)-ß, insulin-like growth factor (IGF), and epidermal growth factor (EGF) production, are stimulated by MSCs [12].

Therefore, we aimed to evaluate the effects of a decellularized omentum scaffold in critical bone defects. In addition, the potential osteoinductive effects of the combination of PRP and MSCs were investigated.

## 2. Materials and Methods

The study was executed after receiving the ethics committee at Hacettepe University (Ankara, Turkey), Medical Faculty Animal Research Laboratory, (Date: 25 July 2017, 2017/49-08).

### 2.1. Mesenchymal Stem Cell Isolation and In Vitro Expansion

The permission was obtained in accordance with the Helsinki Declaration of 2013 and the University of Hacettepe Research Ethics Committee (permit no. 2016/03-01) for the use of mononuclear cells, which were taken from 8-year-old female bone marrow donors, isolated in our archive cryostocks and stored at $-196\,°C$ until use. An amount of $20 \times 10^6$ isolated cells were planted into the tissue culture medium in T-75 flasks with 10 mL of growth medium with 10% fetal bovine serum (DMEM-LG) and 1% penicillin-streptomycin antibiotics. Incubation was performed at $37\,°C$ in 5% $CO_2$. When cells reached 70–80% confluence in 7–12 days, the medium was irrigated with a 0.25% trypsin/EDTA solution for mobilization of non-adherent cells. Then, adhesive cells were cultured in a T-75 flask with a density of $2 \times 10^3$ cells/cm$^2$. In subsequent passages, cells were checked every day with an inverted microscope for non-adherent cells, and the medium changed every three days to remove non-adherent cells. At the 4th passage, adherent cells showed fibroblastic morphology and a homogenous cell population was obtained in all cultures. The cells demonstrated 95% positivity for stromal markers including CD29, CD44, CD73, CD90 and CD105 (e-Biosciences, San Diego, CA, USA). Hematopoietic markers, CD146, CD45 and CD34 (e-Biosciences, USA) were negative. The differentiation capacity of MSCs was confirmed by positive staining in osteogenic and adipogenic differentiation assays after 21 days of culture. A $1 \times 10^6$ cell count for every rat in fetal bovine solution with 0.5 mL volume was prepared [13].

### 2.2. Decellularized Omentum Scaffold Preparation

For the preparation of the decellularized omentum, the omentum was obtained from residues used in previous experiments for different purposes from patients undergoing abdominal surgery for non-tumorous surgeries. Omentum tissue was kept at $-80$ °C and dissolved gradually to $+4$ °C. First, tissues were washed with a PBS solution for 24 h in a 1 L flask. After being washed, tissues were placed for intermittent shaking in a shaker for 24 h to be smashed, then smashed tissues were placed in a 1% ($w/v$) sodium dodecyl sulfate solution and shaken at a constant shaking rate, also for 24 h. The smashed pieces were washed out with PBS solution three times to remove the excess SDS remaining in the tissues. The tissues in PBS were replaced in separate tubes for 48 h in isopropanol solution (%99.9) to remove the lipid tissues. The last part was the enzymatic release of the RNA and DNA, with RNase and DNase (Sigma-Aldrich, Saint Louis, MO, USA) in 37 °C incubation for 16 h. The remaining decellularized omentum parts were washed out with a PBS solution and placed into Petri dishes in a sterile condition as dry state ready for use in surgical procedures [14].

### 2.3. PRP Preparation

Two rats were sacrificed through intracardiac blood aspiration, after an intraperitoneal injection of 50 mg/kg ketamine hydrochloride anesthesia. A sterile, disposable monovette, containing a 3.2% sodium citrate system, was centrifuged at $1800\times g$ rcf for 10 min. After the first centrifugation, two layers were seen in the monovette; the yellow layer consisted of PRP and the red layer of erythrocytes and leucocytes. After separating the two layers, the yellow layer was centrifuged at $4000\times g$ rcf for 10 min. The upper portion of the layer was platelet-poor plasma, and the 1 cm lower layer was PRP. No additional activators were used other than second centrifugation to release the growth factors. The separated part of the lower layer was collected and transferred into an injector to be used for 30 min before starting the surgical procedure.

### 2.4. Animal Preparation and Surgical Procedure

Thirty male, inbred Wistar albino rats weighing over 300 g were included in the study after acclimating the laboratory environment for 10 days. The rats were not involved in any previous experiments and were also screened for any diseases. They were kept in metal cages with access to water and food ad libitum. The rats were maintained in $22 \pm 2$ °C environmental conditions with 12 h of light and 12 h of darkness. The animals were fasted for 24 h before the procedure.

Five groups were randomly formed according to the cages in which the animals were housed:

1. Control group ($n = 6$);
2. Scaffold group ($n = 6$);
3. Scaffold + PRP group ($n = 6$);
4. Scaffold + Mesenchymal stem cell group ($n = 6$);
5. Scaffold + PRP + Mesenchymal stem cell group ($n = 6$).

Before the surgical procedure, the decellularized omentum was cut into 0.5 cm$^3$ pieces and was saturated with 0.5 mL PRP or 0.5 mL MSC or both PRP and MSC; it was outside the surgery site in a Petri dish and waited in room temperature (25 °C) for 5 min to adhere to the scaffold. All rats were anesthetized with a 50 mg/kg dose of intraperitoneal ketamine hydrochloride and 3 mg/kg xylazine combination injection. All rats were monitored by a veterinarian during the surgery. The surgical position of the rats was supine and both forelimbs were shaved. Bone defects sized 0.5 cm$^3$ were created in all rats in both radius bones with a 5 mm Kerrison rongeur. Periosteal elevation and muscle retraction were not performed in any rats. All rats in the same groups received the same treatment in both limbs. The bone defect site was treated according to the following group protocol: group 1, bone defect left empty; group 2, previously prepared decellularized omentum sized 0.5 cm$^3$ was placed on all rats' forelimbs; group 3, previously prepared decellularized omentum

sized 0.5 cm³, saturated with 0.5 mL PRP; group 4, previously prepared decellularized omentum sized 0.5 cm³, saturated with 0.5 mL MSCs; and group 5, previously prepared decellularized omentum sized 0.5 cm³, saturated with both PRP and MSCs 0.5 mL (Figure 1). No fixation method was used for the bone. Wound was closed with non-absorbable sutures. After the surgical procedure, while rats were under anesthesia, 15 mg/kg tramadol was used for postoperative analgesia. Each group of rats was labeled and caged in a separate cage with no restriction of activities.

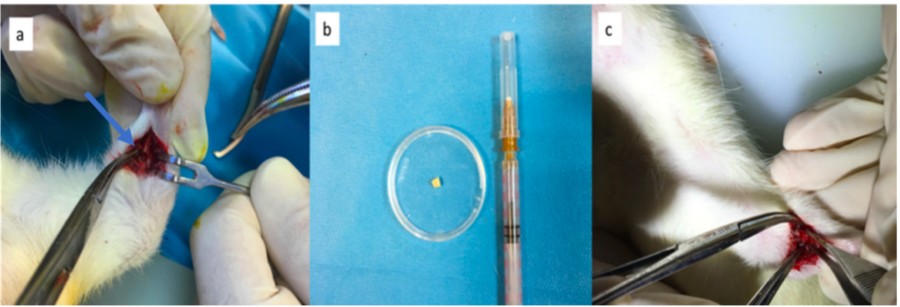

**Figure 1.** Surgical procedure. (**a**) Bone defect. (**b**) Decellularized omentum scaffold with mesenchymal stem cells. (**c**) Bone defect treated with scaffold. * Blue arrow shows the defective area.

## 2.5. Radiologic Analysis

After waiting for 6 weeks of healing, the rats were euthanized, and the forelimbs were disarticulated from the glenohumeral joint in order to obtain anterior-posterior X-rays. Two orthopedic surgeons who were blinded to the groups but were informed about the evaluation method performed the radiologic assessment. Results were scored using the grading scale described by Cook et al. (Table 1) [15].

**Table 1.** Radiographic grading scale for degree of healing (Cook et al.).

| Description | Score |
|---|---|
| No change from immediate postoperative appearance | 0 |
| A slight increase in radiodensity distinguishable from the graft | 1 |
| Recognizable increase in radiodensity, bridging of one cortex with new bone formation from the graft | 2 |
| Bridging of at least one cortex with material of non-uniform radiodensity, early incorporation of the graft suggested by obscurity of graft borders | 3 |
| Defect bridged on both medial and lateral sides with bone of uniform radiodensity, cut ends of the cortex still visible, graft and new bone not easy to differentiate | 4 |
| Same as grade 3, with at least one of four cotices obscured by new bone | 5 |
| Defect bridged by uniform new bone, cut ends of cortex no longer distinguishable, graft no longer visible | 6 |

## 2.6. Histopathologic Analysis

After disarticulation of the forelimbs, the specimens were fixed in a 10% formaldehyde solution for 2 weeks. Then, samples were placed in a 10% ethylenediaminetetraacetic acid solution for the decalcification process. Samples were embedded in paraffin blocks and 5 μm-thick sections were cut through the long axis from the bone defect zone and stained with hematoxylin and eosin (H&E) stain for light microscope analysis. The best sections of the specimens were evaluated by two histolopathologists in the Histology and Embryology Department, who were informed about the histologic grading scale described by Salkeld et al. (Table 2) [16].

**Table 2.** Histological grading scale for the degree of healing (Salkeld et al.).

| Criteria | | Description | Score |
|---|---|---|---|
| Quality of union | | No sign of fibrous or other union | 0 |
| | | Fibrous union | 1 |
| | | Fibrocartilagenous union or cartilage union | 2 |
| | | Mineralizing cartilage and bone union | 3 |
| | | Bone union | 4 |
| Cortex development and remodelling | | No cortex formed | 0 |
| | | Formation of new bone along exterior borders | 1 |
| | | Recognizable formation of both the outer cortex border and the medullary space | 2 |
| | | Coritces formed but incomplete bridging | 3 |
| | | Complete formation of cortices with bridging of defect | 4 |
| Bone-graft incorporation and new bone formation | No new bone, all or most of graft visible | Graft material present, no incorporation, no new bone formation | 0 |
| | | Graft present, some incorporation with new bone formation and small amount of new bone | 1 |
| | | Graft present, some incorporation with new bone formation and moderate amount of new bone | 2 |
| | Decreasing graft, increasing new bone | Graft present, some incorporation with new bone formation continuous with host bone and early remodelling changes in new bone | 3 |
| | | Decreased amount of graft (compared with grade 3) good incorporation of graft and new bone with host and ample new bone | 4 |
| | | Less amount of graft still visible (compared with grade 4), good incorporation of graft and new bone with host and ample new bone | 5 |
| | No graft visible, extensive new bone | Difficult to differantiate graft from new bone, excellent incorporation and advanced remodelling of new bone with graft and host | 6 |

*2.7. Statistical Analysis*

The SPSS software, version 21.0, was used for statistical analysis. Descriptive statistics included median (minimum and maximum) values. The Kruskal-Wallis analysis of variance (ANOVA) was used to compare the groups in terms of histologic, radiologic and biomechanical results. After Kruskal-Wallis ANOVA was performed, the Mann-Whitney U test was performed with Bonferroni correction for paired comparison of groups. Results are expressed with 95% confidence intervals. Significance was defined as $p < 0.05$.

**3. Results**

All 30 rats woke from anesthesia, and no major wounds or other complications occurred. After all rats were euthanized, radiologic and histopathologic studies were performed.

*3.1. Radiologic Findings*

A total of 60 sample assessments were performed according to the Cook scale. In the radiologic assessment, compared to the control group, radiological healing was significantly higher in the omentum and omentum + PRP-treated groups ($p = 0.047$ and $p = 0.047$). In addition, the omentum and omentum + PRP groups had significantly higher healing than the omentum + mesenchymal stem cell group ($p = 0.047$ and $p = 0.047$) (Table 3). Radiologic data are shown in Figure 2.

**Table 3.** Radiological assessment scoring of groups according to Cook criteria.

| Group | N | Median | Minimum | Maximum | *p* |
|---|---|---|---|---|---|
| Control | 12 | 1.0 | 0 | 1 | |
| Omentum | 12 | 1.5 | 1 | 3 | |
| Omentum + MSC | 12 | 1.0 | 0 | 1 | <0.05 |
| Omentum + PRP | 12 | 1.5 | 1 | 3 | |
| Omentum + MSC + PRP | 12 | 1.0 | 0 | 3 | |

* Data are given according to median (minimum-maximum).

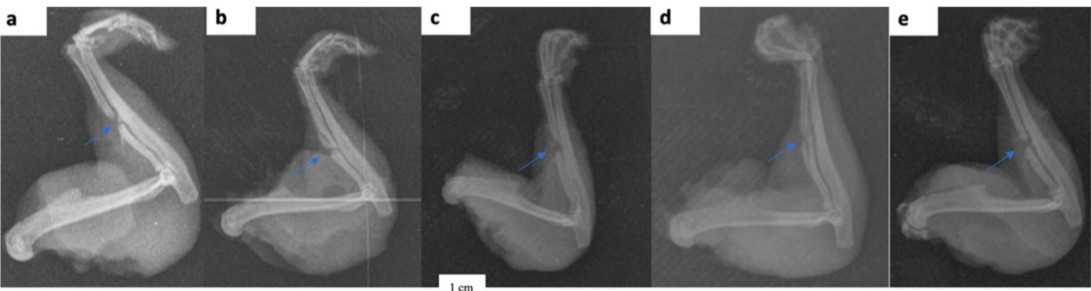

**Figure 2.** Radiologic samples of the groups. (**a**) Control group. (**b**) Omentum group. (**c**) Omentum + Mesenchymal stem cell group. (**d**) Omentum + PRP group. (**e**) Omentum + Mesenchymal stem cell + PRP group. * Blue arrows show the defective area.

### 3.2. Histopathological Findings

Histologically, in the sections, the defect was still visible when looking at the control sample, and there were new bone islands and adipose tissue near the defective areas. In the omentum group specimen, the defect was completely loaded with new cells and the scaffold was organized. In the omentum + mesenchymal stem cell group specimen, the scaffold was still visible, meaning collagen was visible; most of the living cells and newly formed fat cells were in the scaffold with some bone islands and minimal calcification. In the omentum + PRP specimen, a reduction of defect size and increased calcification were observed with a fully organized scaffold with bone cells. In the omentum + mesenchymal stem cell + PRP group, there were increased vascular structures and fat tissue as well as increased bone islands and calcification compared to omentum + mesenchymal stem cell group sample. Histologic sections are shown in Figure 3.

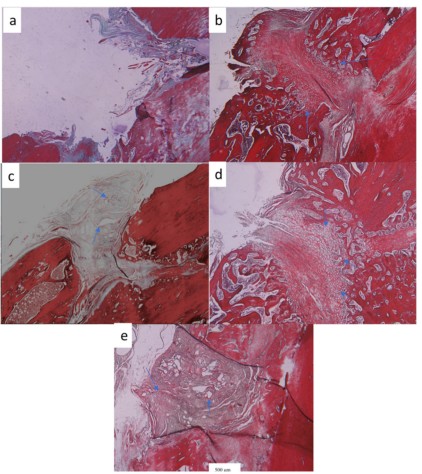

**Figure 3.** Histopathologic sample of each group. (**a**) Control group sample 4× magnification, defect still seen. (**b**) Omentum group sample 4× magnification, defect fully loaded with new cells and scaffold was organized with new bone islands (blue arrows). (**c**) Omentum + mesenchymal stem cell sample 4× magnification, scaffold still seen with new formed fat cells within scaffold (blue arrows)

and minimal calcification. (**d**) Omentum + PRP group sample 4× magnification, with shrinkage of defect size with full organized scaffold, with bone islands (blue arrows). (**e**) Omentum + mesenchymal stem cell + PRP group 4× magnification, increased vascular structures and fat tissue, also increased bone islands and calcification (blue arrows).

Sixty samples were assessed according to the method of Salkeld et al. Evaluation was based on the following parameters: (1) quality of union, (2) cortical development and remodeling and (3) new bone formation.

Between the five groups, there was a significant difference according to the Salkeld scoring system for each parameter and total score. For the quality of the union, the control group was significantly lower than the omentum and omentum + PRP used groups ($p = 0.004$ and $p = 0.016$). In addition, the omentum + mesenchymal stem cell-treated group was significantly lower than omentum and omentum + PRP-treated groups ($p = 0.001$ and $p = 0.006$).

According to the cortical development and remodeling scores, the control group was significantly lower than the omentum and omentum + PRP-treated groups ($p = 0.001$ and $p = 0.001$). In addition, the omentum + mesenchymal stem cell-treated group was significantly lower than the omentum and omentum + PRP-treated groups ($p = 0.001$ and $p = 0.007$).

According to the new bone formation scores, the control group was significantly lower than the omentum and omentum + PRP-treated groups ($p = 0.001$ and $p = 0.001$). In addition, the omentum + mesenchymal stem cell-treated group was significantly lower than the omentum and omentum + PRP-treated groups ($p = 0.013$ and $p = 0.016$).

For total scores, the control group was significantly lower than the omentum and omentum + PRP-treated groups ($p = 0.001$ and $p = 0.001$). In addition, the omentum + mesenchymal stem cell-treated group was significantly lower than the omentum and omentum + PRP-treated groups ($p = 0.002$ and $p = 0.003$) (Table 4).

**Table 4.** Histopathological assessment scoring of groups according to Salkeld et al.

| Group | N | Quality of Union | Cortical Development and Remodelling | New Bone Formation | Total Score | p |
|---|---|---|---|---|---|---|
| Control | 12 | 2 (1–2) | 0 (0–0) | 1 (0–1) | 3 (1–3) | |
| Omentum | 12 | 3 (2–3) | 3 (2–3) | 3 (3–3) | 9 (7–9) | |
| Omentum + Mesenchymal stem cell | 12 | 1.5 (1–2) | 0 (0–1) | 1 (1–2) | 2.5 (2–5) | <0.01 |
| Omentum + PRP | 12 | 3 (2–3) | 2.5 (2–3) | 3.5 (2–4) | 8.5 (6–10) | |
| Omentum + Mesenchymal stem cell + PRP | 12 | 2 (2–3) | 2 (2–3) | 2 (1–3) | 6.5 (3–9) | |

* Data are given according to median (minimum–maximum).

After use of our decellularized omentum scaffold, the vitality of new cells was maintained, new bone formation occurred, the defect size became smaller and vanished, and the scaffold was resorbed. With the combination of MSCs, new bone formation was diminished, and bone healing was damaged. Decellularized omentum with a combination of PRP increased bone healing.

## 4. Discussion

The present study is an experimental animal study which evaluates whether the decellularized omentum could be used as an effective scaffold in critical bone defects. In the previous literature, it is well shown that decellularized scaffolds promote cellular adhesion and migration in addition to cell proliferation and differentiation [17]. We preferred omentum in our study because of its natural basis to increase the biocompatibility. In current literature, many studies show that omentum provides support and growth of

the cells as a bioreactor in soft tissues [18,19]. However, decellularized omentum has never been tried in orthopedic literature as a scaffold. Our study, for the first time, showed that decellularized omentum could be used as an efficient scaffold. We investigated omentum in animal models despite its human origin, but it should be kept in mind that the decellularization process enabled us to lose the antigenic activity of the omentum.

Histologic data showed that ossification was seen, the defect shrank and resorption of the scaffold occurred. In addition, in groups in which the scaffold was used, it was seen that the decellularized omentum scaffold allowed cellular adhesion and proliferation and supported cellular vitality. These results make us think that the omentum is an inspiring scaffold that can be used in the treatment of critical bone defects.

The other goal of the study was to find a way to increase bone healing in addition to a suitable scaffold. PRP was proven to increase bone healing in the literature previously [20]. In addition, we choose to use a previously described double-centrifugation method, as the literature supports that double centrifugation results in greater release of platelets and growth factors [21,22]. In our study, histologically, a PRP combination with omentum had higher organized ossification and bone mineralization than the omentum only group. In addition, the PRP + mesenchymal stem cell and omentum combination group had higher bone islands than the mesenchymal stem cell and omentum combination group. These results show that PRP increases bone union microscopically.

For bone healing, the two methods, MSCs and PRP, were demonstrated to achieve improvement. Even though we prepared allogenic PRP, we could not prepare allogenic MSCs due to technical drawbacks. However, previous animal studies that used human-origin MSCs with or without a suitable scaffold illustrated no systemic or local rejection reactions, which encouraged us to design our study [23,24].

There are various studies regarding the effects of MSC usage in the healing of critical bone defects. In some studies, it was shown that MSC combinations increase osteoinduction and osteogenesis [25,26]. CD44, CD45, CD73, CD90 and CD105 surface markers are essential for using MSC in orthopedic treatments to achieve osteogenic formation [27]. In our study, we used MSCs that express those surface markers. Although the decellularized omentum scaffold allowed cell adhesion and proliferation and the used MSCs had potential for osteogenic transformation, microscopic examinations and Salkeld scoring systems showed decreased bone healing and new bone formation.

There are studies claiming direct injection of MSCs into critical bone defects and non-unions impairs healing [28]. When we looked at the mechanism which may have caused this, it was shown that bolus MSC could lead to a reduction in gene expression of Runt-related transcription factor-2 (Runx2) and Osterix pathway via TNF-$\alpha$, which was responsible for the transformation of MSCs to osteogenic cells [29]. It is known that cumulative MSC expression causes a decrease in TNF-$\alpha$ [30,31]. Scaffold applications with low doses of TNF-$\alpha$ decreased osteogenic activation, and high doses of TNF-$\alpha$ increased osteogenic activation in an experimental rat study [32]. The anti-inflammatory effect of cumulative MSC applications might have decreased TNF-$\alpha$ levels in our scaffold and callus formation. This knowledge led us to speculate that this may be the reason for the impairment of osteogenic formations with MSC applications.

Using xenogenic mesenchymal stem cells could be another reason for decreased healing. Although there are many studies that used xenogenic mesenchymal stem cells that caused no local or systemic rejections, there are also some controversial studies describing human mesenchymal stem cell apoptosis and fragmentation in immune-competent mice [33]. It has been reported that xenogeneic, MSC-derived chondrocytes trigger T lymphocyte proliferation, cytotoxicity, increasing antigen presentation and further activation of the adaptive immune response [34]. The possibility of localized immune reaction and diminishing mesenchymal stem cells may also explain impairment of osteogenic formation [35].

Radiologic results showed a correlation with our histologic results. We think that these correlations empower our study. However, our scaffold was not fully ossified and

radiologic healing was not completely seen. There is no similar study in the literature to discuss how much time would be needed for radiological healing.

One of the limitations of our study is the limited number of samples. Another limitation is that, although PRP is allogenic, MSCs are not. Due to technical drawbacks, rat-origin MSCs could not be used in our study. In addition, we did not investigate the important features of our scaffold, such as porosity and micro and nano construct.

## 5. Conclusions

This research demonstrates that a decellularized omentum scaffold is suitable for critical-sized bone defects, and similar results in in vivo tests and biomechanical studies highlight the usage of decellularized omentum as a scaffold in clinical practice for segmental bone defects. With the advancement of storage conditions, pre-prepared omentum grafts can be used in planned or emergency cases.

**Author Contributions:** A.E. made the animal preparation and surgical procedure, and analyzed and interpreted the data. E.O. and E.T. performed the measurements and scorings of radiologic data. A.C.S.Y. and R.H. performed the histologic data analysis. D.U.C. and E.K. prepared the mesenchymal stem cells. A.E. and E.T. reviewed the manuscript and planned the research. All authors have read and agreed to the published version of the manuscript.

**Funding:** This research did not receive any specific grant from funding agencies in the public, commercial or not-for-profit sectors.

**Institutional Review Board Statement:** The study was conducted according to the guidelines of the Declaration of Helsinki and approved by the Institutional Review Board (or Ethics Committee) of Hacettepe University Medical Faculty Animal Research Laboratory (Date: 25 July 2017, 2017/49-08).

**Informed Consent Statement:** Informed consent was obtained from all subjects involved in the study. Written informed consent has been obtained from the patients to publish this paper.

**Data Availability Statement:** The data used and/or analyzed during the current study are available from the corresponding author on reasonable request.

**Acknowledgments:** The authors thank the Tissue Laboratory for decellularized omentum production and Stem Cell Laboratory workers for their high efforts.

**Conflicts of Interest:** The authors declare no conflict of interest.

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
