# Peer review of "Effect of a Decellularized Omentum Scaffold with Combination of Mesenchymal Stem Cells and Platelet-Rich Plasma on Healing of Critical-Sized Bone Defect: A Rat Model"

_applsci, doi:10.3390/app112210900_

Round 1
Reviewer 1 Report
- The introduction part is greatly lacking with scientific content in regard to omentum scaffolds. When surveyed in the literature, I would find plenty of articles with respect to omentum based scaffolds in tissue engineering. Therefore, I request the authors to strength the introduction part by citing all important articles and make a hypothesis why your manuscript is different from others. Moreover, authors must emphasis on the novelty aspect of this paper.
- Though the authors have used SDS treatment for decellularization, there are no results in regard to validate the effect of decellularization in the scaffolds. Therefore, I request the authors to add some fluorescent images or some graphs to show the decrease in cellularity in the scaffolds post SDS treatment
- Usually PRP needs to be activated by either CaCl2/Thrombin/cycling freeze thawing in order to release the growth factors that can improve the healing process. The authors have not mentioned about any activators in this paper. If no activators are used then due justification is needed.
- The final form of omentum scaffold prior implantation needs to be mentioned. In interest of the audience, it is essential to mention how the scaffold was fabricated and whether it was in dry/wet state or freeze drying process was employed
- The radiographic images are very poor in quality, I request the authors to enhance the quality of image and necessary arrow marks are needed to represent the better healing qualities in the best performing groups
- Statistical analysis is missing in Figure 4
- In H&E staining indicate the presence of tissue vascularization, neo tissue formation and collagen deposition using arrow marks

Author Response
Dear Associate Editor and Reviewers,
Thank you very much for your valuable comments. Your comments were insightful and helped us to improve our manuscript.
Yours sincerely,
Abdulsamet Emet

Reviewer 2 Report
Authors aimed to study the effects of a decellularized omentum based scaffold in combination with platelet-rich plasma (PRP) and mesenchymcal stem cells (MSCs) in a rat radius critical defect model. Through radiologic and histopathological data, omentum scaffold, along with PRP significantly increased bone formation and healing. The addition of mesenchymal stem cells was observed to be detrimental in bone formation and healing. Authors attribute possible anti-inflammatory effects of cumulative MSCs, which may have contributed to decreased gene expression of Runx2 and TNF-α, or the use of xenogenic MSCs. Overall, authors showed the potential use of deceullarized omentum scaffold for bone healing.
Detailed Suggestions:
- Please consider revising the statement from line 35, “Critical bone defects are difficult situation and since, when self-healing is not expected despite enough surgical stabilization”.
- Please include in the Materials and Methods Section, how it was decided which two treatments individual animals received, since both forelimbs were operated on. Did a single animal receive the same treatment in both limbs?
- For histopathologic samples, please include the sample number per group.
- Please be consistent in how groups are referred. For example, for the omentum and PRP groups, are later referred to as omentum + PRP. Consider keeping the “+” instead of “and” for easier readability.
- Please indicate where the defect is in Figure 2.
- Please indicate what the blue arrows in Figure 3 represent.
- Please increase text size of axis labels in Figure 4.
- Please cite references after sentence in line 280, “There are studies claiming direct injection of MSCs into critical bone defect and non-unions impairs healing.”
- In discussion, consider including how scaffold features such as porosity and topography may influence scaffold performance.
Author Response

(The authors gave the same response as above.)

Reviewer 3 Report
Dr. Emet and colleagues have submitted a manuscript entitled: " Effect of a Decellularized Omentum Scaffold with Combination of Mesenchymal Stem Cells and Platelet Rich Plasma on Healing of Critical-sized Bone Defect: A Rat Model." The study in an in vivo animal model comparing omentum scaffold with MSC and/or PRP to control group. The authors found that omentum alone and omentum plus PRP promoted a greater degree of bone healing. The authors conclude the use of decellularized omentum scaffold is suitable in the healing for critical bone defects .
Materials and Methods
Line: 85 – 89: Provide a supplementary figure of FACS analysis
Line 123: Describe the randomization method
Line 132: Please, confirm if you used ketamine alone or in combination with xylazine
Lines 137 – 143: Describe the conditions used for omentum saturation with PRP and cells (time? Temperature?...)
Table 1: line 4 spell check
Table 2: spell check lines 10 and 12
Results
Figure 2: Change position of Figure 2b
Table 3: Indicate differences among groups. Suggestion: use MSC for the cells and provide abbreviatures in the footnote
The histopathological findings need more attention. It is not possible to see “new bone areas”in the control group. To confirm the “increase in vascular structures”authors should provide a higher magnification image or perform an IHC for endothelial cells.
Figure 3: You must include scale bar in all images. Figure legend: blue color or blue arrow?
Line 213: Please, explain why only 40 samples were assessed
A descripotion of the results of omentum+MSC+PRP group is missing in all aspects evaluated.
Figure 4 in unnecessary once is presenting same data of Table 4.
Discussion
The authors should discuss how their scaffold could be included into clinical practice.
Author Response

(The authors gave the same response as above.)

Round 2
Reviewer 1 Report
The release of growth factors from PRP cannot be achieved without any activators. Because without any activators the rupture of alpha granules does not occurs and this eventually impedes the release of growth factors. If authors propose that two-step centrifugation is more sufficient to release the growth factors then this claim needs to be supported by stating and quoting similar studies in the discussion part.
Author Response

(The authors gave the same response as above.)
